# The Impact of Protein and Amino Acid Supplementation on Muscular Strength and Endurance in Recreational Gym-Goers During 8-Week Resistance Training

**DOI:** 10.3390/sports13060182

**Published:** 2025-06-11

**Authors:** Sandor-Richard Nagy, Magdalena Mititelu, Ruxandra-Cristina Marin, Violeta Popovici, Annamaria Pallag, Tünde Jurca

**Affiliations:** 1Doctoral School of Biomedical Sciences, Faculty of Medicine and Pharmacy, University of Oradea, 410073 Oradea, Romania; nagy.sandorrichard@student.uoradea.ro (S.-R.N.); apallag@uoradea.ro (A.P.); tjurca@uoradea.ro (T.J.); 2International Fitness and Bodybuilding Federation, 28232 Madrid, Spain; 3Department of Clinical Laboratory and Food Safety, Faculty of Pharmacy, “Carol Davila” University of Medicine and Pharmacy, 020956 Bucharest, Romania; magdalena.mititelu@umfcd.ro; 4Center for Mountain Economics, “Costin C. Kiriţescu” National Institute of Economic Research, Romanian Academy, 725700 Vatra-Dornei, Romania; 5Department of Pharmacy, Faculty of Medicine and Pharmacy, University of Oradea, 410073 Oradea, Romania

**Keywords:** whey protein, creatine, L-carnitine, recreational gym-goers, resistance training, one repetition maximum

## Abstract

**Objective:** Most recreational gym-goers independently consume nutritional supplements (NSs) without physician advice and a personalized diet. The present study examines the preference for nutritional supplements (NSs) based on protein and amino acids of 218 recreational gym-goers (males and females aged 18–60). It also investigates the NS’s impact on resistance training (RT) performance. **Methods:** All participants (*n* = 218) were regular members of two gym centers in Oradea. Baseline data and information about daily diet and supplement preferences were obtained through face-to-face interviews. At the same time, RT performance was assessed by measuring 1RM in six exercises three times (W0, W4, and W8). **Results:** Our findings reveal that 24.3% of participants did not consume NSs, while the majority (75.6%, *p* < 0.05) used them to improve their physical condition; men were more likely to consume NSs than women (83.3% vs. 63.9%, *p* < 0.05). Gym-goers were grouped based on their NS consumption: L-carnitine, creatine, whey protein (WP), and triple combination; the non-supplemented group was the control. The combination substantially correlated with a balanced diet, 3001–3500 and >3500 calories/day; creatine was appreciably associated with 2001–2500 calories/day; L-carnitine was associated with 151–200 g protein/day, while control was considerably linked with a vegetarian diet and <1000 calorie/day (r > 0.900, *p* < 0.05). The results showed that almost all participants exhibited progressive muscle strength improvements. As an overview, 1RM substantially varied with NS consumed, body weight status, and gender (*p* < 0.0001), except for the pull-up count, which varied with NS and gender (*p* < 0.0001). Additionally, 1RM significantly varied with age (deadlift and pull-ups), daily protein consumption (back squats, biceps, and triceps), daily calories (back squats), and diet type (biceps, triceps, and pull-up exercises), *p* < 0.05. On the other hand, most NSs associated with RT exercises led to a general increase in body weight. Only L-carnitine decreased it. **Conclusions:** Resistance training records of recreational athletes are significantly influenced by age, gender, body weight status, NS type, and daily diet features (*p* < 0.05). Our findings highlight the essential role of professional guidance in nutritional supplementation associated with a suitable diet for optimal RT performance of recreational athletes.

## 1. Introduction

Recreational gym practice enhances the quality of life, improving widely aged people’s physical, mental, and social well-being [1,2]. Regular physical activity, encompassing any body movement that expends energy, helps manage weight, builds and maintains strong muscles and bones, increases energy levels, improves mood, diminishes stress, enhances cognitive function, facilitates recovery after various diseases, reduces the risk of chronic ones and ameliorates patient condition in various chronic illnesses [3]. Therefore, numerous people embrace a lifestyle incorporating a balanced diet and regular exercise as a key to a healthier life. To meet nutritional demands and improve performances, athletes and recreational gym-goers use various types of protein powders, amino acids, and vitamins [4,5,6]. The prevalence of NS use varies widely, ranging from 40% to 100%, depending on the type of sport, the level of competition, and personal goals [7]. Various protein and amino acid supplements are widely marketed to athletes and recreational gym consumers to promote muscle growth and enhance sports performance. Creatine and protein are in the top five of the most used NSs [8]. Frequently consumed by gym-goers are protein powders (59.17%), followed by creatine (41.28%), and L-carnitine (5.05%) [9]. Most recreationists prefer protein associated with creatine and amino acids (48.8%) or protein and creatine alone (6.4%) [10]. Protein supplements, particularly whey protein (WP), are among the most popular due to their role in muscle synthesis and recovery [11]. The Institute of Medicine’s recommended dietary allowance (RDA) for protein is 0.8 g/kg/day for the entire adult population [12], while the Acceptable Macronutrient Distribution Range (AMDR) indicates a daily protein intake of 1.05–3.67 g per kilogram of body weight for adults aged 18 years and older [13]. Whey protein (WP) has had a significant impact on nutritional supplements for the community, particularly athletes, as it contains around 50% of essential amino acids (EAAs) and approximately 26% of branched-chain amino acids (BCAAs). The scientific literature provides extensive evidence that protein and isolated amino acid supplementation enhances muscle growth and recovery after exercise and adaptations to resistance training.

Creatine is a well-documented ergogenic aid that facilitates ATP production by enhancing muscle strength, lean body mass, and recovery [14]. Creatine monohydrate (CM) supplementation has been consistently reported in the literature to increase phosphagen levels in muscle, improve performance during repetitive high-intensity exercise, and promote significant training adaptations [15]. It is a stable form of creatine that is not significantly degraded during the digestive process and is either taken up by muscle or eliminated in the urine. Despite its widespread use worldwide, no clinically significant adverse effects have been reported from CM supplementation; short- and long-term supplementation is safe and well-tolerated in healthy individuals and several patient populations. The regulatory status of CM is not well-established; currently, it is the only form of creatine officially approved in key markets, including the USA, Canada, the European Union, and South Korea [15]. Research has shown the effects of creatine supplementation, including injury prevention and rehabilitation, enhanced post-exercise recovery, increases in serum testosterone concentration, and reduction in cortisol level [16], as well as potential neurological benefits relevant to sports [17,18]. Supplementation protocols include an initial loading phase (0.3 g/kg/day) for 5–6 days, followed by a maintenance dose that varies in different studies: 0.03 g/kg/day [19], 0.07 g/kg/day [20] or 0.1 g/kg/day [21]. Previous studies regarding creatine supplementation’s effects on RT performance were conducted both ways, with and without a loading protocol [22,23].

L-carnitine, another widely used supplement, is valued for facilitating the metabolism of fatty acids and energy production within mitochondria [24]. Increasing L-carnitine intake through supplementation can enhance fat oxidation, significantly reducing body fat reserves and facilitating weight loss [25]. This is the main reason for L-creatine inclusion in recreational gym-goers’ preferences. On the other hand, 1–2 g L-carnitine/day increases exercise performance, improves recovery, and reduces oxidative stress [26,27]. Although naturally present in animal-based foods, its supplementation is often necessary for individuals with higher metabolic demands or specific treatments and dietary restrictions [28,29,30,31,32,33,34,35,36,37,38].

While the use and benefits of NSs in professional athletes have been extensively studied, data about their consumption by recreationists are still limited [39]. A recent study shows that NSs are used by 60.6% of athletes and 67.3% of recreationists [40]. Other authors reported a higher NS consumption in recreational vs. professional athletes (62% vs. 50%) [41]. Most recreationists lack knowledge about daily diet’s influence on sports performance, especially significant aspects (e.g., protein sources); despite this, they make their own decisions, raising NS consumption [42].

In this context, our study examines nutritional supplement preferences in recreational gym-goers, including males and females aged 18–60; it investigates the effects of different protein and amino acid supplements (whey protein, creatine, L-carnitine, and their combination) on RT records across six common exercises.

The following hypotheses underline the present study:Recreational gym-goers commonly use protein and amino acid supplements alone or in combination [10,43,44];Most of them are young men aged 18–30 [42,45,46,47,48];Age, sex, daily diet type, and NS type significantly influence gym-goers’ RT records [6,49,50,51,52,53].

Our findings offer critical insights into these NSs’ influence on resistance training performance across various exercises, investigating potential differences associated with gender, age, diet type, and body weight. Previous studies have often examined isolated supplements or single exercises on low-size and relatively uniform groups (with individuals from restricted age groups, only males or females with similar body records, and controlled diets), lacking a comprehensive approach considering their combined effects on various resistance exercises. A significant gap addressed in the present study is the limited data on how self-selected protein and amino acid supplements influence RT performance in a heterogeneous recreationist group across six exercises over an 8-week training. Additionally, this study fills the gap related to gender-, age-, and body weight-specific responses to these NSs, which are often overlooked in sports nutrition research. By evaluating variations in RT performance among men and women and across different age and BMI groups, our study provides valuable data for tailoring protein and amino-acid supplementation and training programs to individual needs. Furthermore, our findings underscore the importance of professional guidance in nutritional supplementation associated with a suitable diet for optimal RT performance.

## 2. Materials and Methods

The present research was conducted with approval from the Ethical Committee of the Faculty of Medicine and Pharmacy at the University of Oradea, Romania, No. 21/25.02.2021, and from the Gym’s Management Committee, No. 7/03.10.2022.

It involves 218 gym-goers who regularly exercise at two popular gyms in Oradea, Romania. The distance between the home and the gym location was the reason for the recreational gym-goers’ distribution between the two gym centers. The study was simultaneously conducted in both gyms, where two qualified trainers supervised the NS administration and monitored the RT protocol.

### 2.1. Sample Size

Recently, it was estimated that over 800,000 Romanians are gym members, according to https://www.gsp.ro/sporturi/fitness/cifre-alarmante-pentru-romania-activitati-fizice-sport-687913.html, accessed on 20 May 2025. The sample size was calculated considering an error percentage of 6%, a confidence interval of 95%, and a response rate of 95%. Only 218 of the 230 gym-goers invited to the present study completed it, representing 0.027% of the population’s size.

### 2.2. Inclusion and Exclusion Criteria

According to the inclusion criteria, two qualified coaches rigorously checked and selected potential participants at both gym locations, considering nutritional supplementation and RT practice duration. Healthy males and females aged 18–60 were chosen from regular members of the gymnasiums mentioned above. They should have practiced RT exercises for at least 6 months and have a self-supplementation history of at least 3 months with whey protein, L-carnitine, or creatine (alone or in triple combination). A control group of no NS consumers was formed.

Individuals with severe illnesses (musculoskeletal disorders, cancer, liver disease, heart failure, or kidney failure), those who used androgenic steroids, diuretics, epinephrine, and other prohibited substances in the gym that are interdicted by the World Anti-Doping Agency (WADA) [54,55] were excluded from the study. Another exclusion criterion refers to individuals who regularly performed high-intensity exercises for gym competitions and those who commonly performed their gym training in other gym centers.

### 2.3. Essential Considerations

Baseline data were collected using a face-to-face questionnaire. Each participant signed a written informed consent form to ensure voluntary participation and suitable adherence to the study protocol.

All participants (*n* = 218) were members of one of the previously mentioned gym centers. Data about supplement preferences were obtained using a face-to-face interview based on the questionnaire adapted from [45,56]. The participants provided data regarding their diet, including the diet type, daily protein, and calorie intake, and mentioned the preferred nutritional supplement.

They were divided into four groups: WP users (n = 42), creatine supplement users (n = 38), L-carnitine supplement users (n = 37), and triple combination users (n = 48). No NS consumers for at least three months formed the control group for all supplemented participants (n = 53).

All participants received brief instructions on the study’s purpose, and all data collected were registered and centralized in Excel.

The gym protocol consists of 6 RT exercises: back squats, bench presses, deadlifts, biceps curls, triceps extensions, and pull-ups (Figure 1).

### 2.4. Study Design

All participants were engaged in a resistance training protocol for eight weeks, including administering NS and performing the same exercises. They were grouped in pairs and ruled in three training sessions per week. Two qualified sports coaches, one in each gymnasium, monitored the administration of NS, ensuring their consumption as illustrated in Appendix A, and supervised the training sessions.

The study protocol included 3 essential measurement points: at the beginning, in the first meeting (W0), after 4 weeks (W4), and after 8 weeks (W8).

Measurements of body weight and height were initially performed (in W0) for all individuals, and the body mass index (BMI) was calculated under WHO guidelines using the Quetelet equation: body weight (kg)/height^2^ (m) [57]. After a warm-up, an initial RT session aims to measure muscular strength through a one-repetition maximum (one-rep max, 1RM), representing the highest weight that can be lifted successfully through a full range of motion [58].

In W4 and W8, body weight and 1RM values were measured for each exercise.

### 2.5. Experimental Approach

#### 2.5.1. Nutritional Supplementation Protocol

The coaches suggested high-quality nutritional supplement formulations according to individual preferences (Appendix A) and established the mode of administration. Thirty minutes before training, each participant received creatine (1 dose = 3000 mg) and L-carnitine (1 dose = 2000 mg). After training, they received WP (30 g WP containing 25 g protein isolate). Male gym-goers received 1 WP dose more than females; the second dose was consumed in the morning, after breakfast.

The dosage of whey protein differs between women and men due to physiological differences. Men have greater muscle mass and body weight, which implies a higher protein requirement. In addition, testosterone—more present in men—enhances protein synthesis, allowing them to use higher doses (40–50 g vs. 20–25 g in women) efficiently. Thus, the doubling of the dose is not arbitrary but proportional to the men’s body requirements. The second WP dose was taken in the morning because, after an overnight fast, the body is in a catabolic state, in which it stops muscle degradation and reactivates anabolic processes. Even if the RT session is scheduled later, this WP morning dose contributes to a balanced distribution of proteins during the day—a recommended strategy for maximizing muscle recovery and lean mass growth. It can also correct breakfast, which is often lacking in proteins.

The NSs were administered exclusively on gym days, before or after training, at each gymnasium (Appendix A); on the same days, WP user males also had to consume the first whey protein sachet in the morning. Combination users (males and females) consumed all 3 supplements in the same daily doses, as recorded in Appendix A.

Through a personal statement, each participant confirmed adherence to the NS type, formulation, brand, dose, and mode of administration.

Most supplements were administered at the gym by the coaches. Only the morning WP dose administration was confirmed by each male participant on their own responsibility.

#### 2.5.2. Training Protocol

Each training session consists of 6 exercises performed at 5 sets of repetitions with a 60 s resting interval between sets and lasts approximately 70 min. Each week contains 3 RT sessions, and the weekly training loads of all participants increased by 2–10%, depending on the strength of each gym-goer [59].

#### 2.5.3. Warming Protocol

The RT exercise type influences the prior light warm-up protocol, specifically for all muscles involved in each exercise. Therefore, before the biceps, triceps, pull-ups, and bench press exercises, participants took a 10 min warm-up walk on a flat treadmill (0% incline) at a 4 km/h speed [60]. Before starting the deadlift and back squat, participants had a 10 min stationary cycling warm-up at an individually selected intensity [61].

#### 2.5.4. Direct 1RM Testing Protocol (The Gold Standard)

Direct 1RM testing consists of a single repetition with the highest weight possible for an RT exercise. This method, the most accurate way to determine 1RM, was applied step by step, following the Agency for Clinical Innovation Guide, v.2022 [62].

All participants should have been strength-specific and underwent a one-repetition maximum (1RM) load program for each exercise. The coaches checked the participants’ performance to ensure they performed the movements correctly. The 1RM measurements were performed in W0, W4, and W8, and two RT exercises/day were planned for Monday, Wednesday, and Friday.

After the participants’ 1RM initial values (W0) were determined, the training loads were calculated separately for each participant. The 1RM increased every four weeks (in W4 and, respectively, W8) based on the muscle strength gained by each gym-goer [63]. Most performances were recorded in mass units (kg), except for pull-ups, which were counted. All data were recorded for each gym-goer in Excel (Microsoft 365 2025, v. 18.2503.1198.0).

### 2.6. Statistical Analysis

Statistical analysis was conducted using XLSTAT Premium 2025 (v. 2025.1.0.1427, Lumivero, Denver, CO, USA). All recorded data were processed using Descriptive Statistics to investigate the dataset’s central tendency, dispersion, and distribution, providing essential insights into the study variables [64]. The ANOVA test was used to compare variances between two samples, with *p*-values less than 0.05 considered statistically significant. Multiple samples were compared with the Kruskal–Wallis test [65]. Fischer’s F-test was used to characterize the continuous variable 1RM with baseline categorical variables. Principal Component Analysis correlated NS consumption with various baseline data. The impact of NS consumption, diet type, sex, and age on RT performances and the influence of NS consumption and RT on body weight were also investigated. Finally, the MANOVA test was performed to predict the impact of all baseline data and NS consumption on RT performance and body weight measurements.

## 3. Results

### 3.1. Sociodemographic and Baseline Characteristics of Participants

The study enrolled 218 participants: 86/128 women (36.95%) and 132 (60.55%) men. Regarding age groups, 113/218 (51.83%) participants were from the 18 to 30 age group, 98/218 (44.95%) were between 31 and 50, and 7/218 (3.21%) were aged 51–60 (Table 1).

Of the total participants, 51.3% (112/218) were NW, 65/128 (75.5%) of women, 47/128 (35.6%) of men), 0.9% (2/218) were obese, 44.9% (98/218) were OW (18.6% of women, 62.1% of men), and 2.7% (6/218) were UW.

Most of them had a balanced diet (130/218, 59.63%), 46/218 (21.10%) preferred a hyperprotein diet, 27/218 (12.39%) were vegetarians, and 15/218 (6.88%) had a low-carb diet.

Our study reveals that 24.31% of participants did not consume NS, while the majority (75.69%, *p* < 0.05) used them to improve their physical condition (Table 1). Men were more likely to consume NS than women (83.33% vs. 63.95%, *p* < 0.05).

No NS consumption was significantly correlated with a vegetarian diet (r = 0.953, *p* < 0.05), while combination strongly correlated with a balanced diet (r = 0.966, *p* < 0.05, Figure 2A). L-carnitine was considerably associated with obese gym-goers and 151–200 g protein/day (r = 0.999, r = 0.930, *p* < 0.05, Figure 2A,B). Moreover, the combination substantially correlated with 3001–3500 and >3500 calories/day (r = 0.975, r = 0.999, *p* < 0.05), creatine was appreciably associated with 2001–2500 calories/day (r = 0.970, *p* < 0.05), while <1000 calories/day highly correlated with no NS consumption (r = 0.999, *p* < 0.05, Figure 2C).

### 3.2. Nutritional Supplements Consumption

Among NS users, the most popular choices included combination (22%), WP (19.2%), creatine (17.4%), and L-carnitine (16.9%). Women tended to avoid NS consumption more frequently (34.4%), while similar percentages (24.42% and 22.2%) consumed WP and L-carnitine. In men, 17.2% did not use NS, while a significant proportion took WP (22.2%) or creatine (14%).

### 3.3. RT Performance Records

All data are recorded in Table 2. Each 1RM value shows a wide variation range and a high standard deviation. Therefore, no statistical differences exist among the three 1RM measurements. The highest maximal values of 1RM were recorded in the deadlifts (140–190), back squats (140–160), and bench press (120–150), while minimal ones were (10–15). The records for biceps and triceps were almost similar (from 10 to 60–85 vs. 5–10 to 60–90). Pull-ups increase from 0 to 15–38. These RT records substantially vary with NS consumed, body weight status, and gender (*p* < 0.0001), except for the pull-up count, which varies with NS and gender (*p* < 0.0001). Additionally, 1RM significantly varies with age (deadlift and pull-ups), daily protein consumption (back squats, biceps, and triceps), daily calories (back squats), and diet type (biceps, triceps, and pull-ups exercises), *p* < 0.05 (Table 2).

### 3.4. The Impact of Nutritional Supplements Consumption and Resistance Training on Body Weight

Most NSs associated with RT exercises led to a general increase in body weight (Table 3). Only L-carnitine decreased it. However, no NS consumers (control group) can lose weight, depending on diet and RT performance. In the whole group, statistically significant differences were noted between NW vs. OM and obese gym-goers.

### 3.5. NS Influence on RT Performance of Gym-Goers with Various Daily Diet Types

#### 3.5.1. Balanced Diet

All NS consumed by participants with a balanced diet induced significant differences in RT exercises compared to controls (Figure 3A,D,F,I–K, red color). Appreciable differences (*p* < 0.05) can also be revealed between two NSs. The combination contributed to the highest performance, followed by L-carnitine (in back squats, bench presses, deadlifts, and pull-ups, Figure 3A,D,F,K) and creatine (in biceps and triceps exercises, Figure 3 I,J).

In back squat exercises, all NS led to 1RM values significantly different than control in the following decreasing order: combination, L-carnitine, creatine, and WP—103 ± 26, 87 ± 31, 75 ± 23, 68 ± 24 vs. 38 ± 15 (*p* < 0.05, Figure 3A). Moreover, significant differences can be reported between the combination and creatine and WP (103 ± 26 vs. 75 ± 23 and 68 ± 24, *p* < 0.05, Figure 3A).

In bench press exercises, all NSs were efficient, determining 1RM values significantly different than controls: combination, L-carnitine, creatine, WP: 86 ± 25, 60 ± 27, 73 ± 31, 52 ± 32 vs. 24 ± 14, *p* < 0.05 (Figure 3D).

The same trend is available in the 1RM deadlift values: 98 ± 23, 76 ± 26, 88 ± 42, 64 ± 24 vs. 42 ± 13, *p* < 0.05 (Figure 3F). Significant differences are also between combination and WP: 98 ± 23 vs. 64 ± 24, *p* < 0.05 (Figure 3F).

Substantial differences can be observed in pull-ups regarding NS consumption vs. control: 13 ± 4, 9 ± 4, 9 ± 8, 8 ± 6 vs. 2 ± 2, *p* < 0.05 (Figure 3K). Moreover, statistically significant differences occurred between combination vs. creatine and WP: 13 ± 4 vs. 9 ± 4 and 8 ± 6, *p* < 0.05 (Figure 3K).

In the exercises for the biceps and triceps, only 1RM induced by combination, creatine, and L-carnitine are significantly higher than control: 55 ± 15, 46 ± 16, 45 ± 11 vs. 27 ± 12, *p* < 0.05 and, respectively, 64 ± 14, 55 ± 14, 51 ± 14 vs. 34 ± 11, *p* < 0.05 (Figure 3I,J). At the same time, the combination acts significantly higher than WP: 55 ± 15 vs. 37 ± 14, *p* < 0.05, and 64 ± 14 vs. 44 ± 16, *p* < 0.05 (Figure 3I,J).

#### 3.5.2. Hyperprotein, Vegetarian and Low-Carb Diets

Only back squats, bench presses, and deadlifts display significant differences in the hyperprotein diet (Figure 3B,E,G, mixed colors). Combination and L-carnitine act significantly higher than control in back squats exercise: 120 ± 28, 86 ± 20 vs. 55 ± 22, *p* < 0.05 (Figure 3B). Moreover, combination benefits are substantially higher than creatine: 120 ± 28 vs. 63 ± 19, *p* < 0.05 (Figure 3B). In bench press and deadlift, combination vs. control are the only statistical differences: 89 ± 31 and 116 ± 23 vs. 44 ± 20 and 54 ± 19, *p* < 0.05 (Figure 3E,G).

Nutritional supplements significantly impact gym-goers with a vegetarian diet in back squats and deadlifts (Figure 3C,H, green color). The combination acts considerably higher than the control in the back squat: 118 ± 55 vs. 50 ± 23, *p* < 0.05 (Figure 3D). Moreover, L-carnitine has the lowest impact on deadlift 1RM, significantly different from combination: 50 ± 16 vs. 122 ± 41, *p* < 0.05 (Figure 3C).

A low-carb diet associated with a combination significantly increases the pull-up count compared to control: 12 ± 5 vs. 2 ± 1, *p* < 0.05 (Figure 3L, purple color).

### 3.6. NS Influence on RT Performance of Males and Females Gym-Goers of Different Age Groups

Age-wise, younger participants (18–29) have the highest NS intake, with 32.17% preferring combination NS consumption declines with age, with those over 50 showing the lowest intake (Table 4). Therefore, the impact of NSs on gym performance was analyzed separately for males and females across three age groups: young adults (18–30), early middle-aged and middle-aged adults (31–50), and late middle-aged (51–60) adapted from [66].

Generally, no significant differences were recorded between W0, W4, and W8, except for the combination in 1RM bench press values from W0 and W8 (65 ± 17 vs. 95 ± 19, *p* < 0.05) in males. Therefore, we analyzed the final measurement values of 1RM for each RT exercise and focused on significant changes induced by the used NSs.

#### 3.6.1. Age Group 18–30

Substantial differences were recorded in all RT exercises in males (Figure 4A,C–F,H, blue color) and only in three ones (back squat, triceps, and pull-ups) in females (Figure 4B,G,I, mixed colors).

Combination consumer males aged 18 to 30 recorded the most significant results. L-carnitine users ranked second in the back squat and bench press, while creatine users placed second in the biceps and triceps. Significant differences in 1RM back squat compared to the control group were observed in males supplemented with combination and L-carnitine: 113 ± 25 and 99 ± 21 versus 61 ± 17, *p* < 0.05 (Figure 4A). Moreover, the combination effect is considerably higher than WP (85 ± 16, *p* < 0.05, Figure 4A). It also induced significant female results: 60± 21 vs. 30 ± 10, *p* < 0.05 (Figure 4B).

The effects of combination and L-carnitine were significantly higher than control in the bench press: 95 ± 19 and 83 ± 21 vs. 48 ± 16, *p* < 0.05 (Figure 4C).

In the male deadlift, three NSs (combination, L-carnitine, and creatine) resulted in significantly higher performance compared to the control: 109 ± 21, 102 ± 41, and 100 ± 17 vs. 56 ± 14, *p* < 0.05 (Figure 4D). WP consumers performed markedly lower than those using the combination (56 ± 14, *p* < 0.05, Figure 4D).

In the biceps exercise, the combination and creatine demonstrated significantly more potent effects than the control: 58 ± 13 and 57 ± 12 vs. 37 ± 11, *p* < 0.05 (Figure 4E). The combination’s effects are also remarkably greater in the triceps compared to control, WP, and L-carnitine: 68 ± 13 vs. 47 ± 10, 56 ± 9, and 54 ± 11, *p* < 0.05 (Figure 4F). For females, creatine records significantly differ from control: 48 ± 11 vs. 28 ± 8, *p* < 0.05 (Figure 4G).

The combination induced substantially higher pull-up performance than the control: 14 ± 3 vs. 4 ± 3, *p* < 0.05 (Figure 4H). In the female group, creatine and WP led to the highest records and increased performance: 6 ± 2 vs. 2 ± 2, *p* < 0.05 (Figure 4I).

#### 3.6.2. Age-Group 31–50

Significant differences were recorded in all RT exercises between males (Figure 5A,C,E,G,I,K, blue color) and females (Figure 5B,D,F,H,J,L, mixed colors) from this age group.

In male back squats, the highest 1RM value was obtained with the combination (109 ± 30), followed by WP (96 ± 12), both considerably different for control (69 ± 17, *p* < 0.05, Figure 5A). In female back squats, the combination also led to the highest records, followed by WP and creatine, which significantly differ from the control: 79 ± 8, 55 ± 10, and 54 ± 7 vs. 32 ± 11, *p* < 0.05 (Figure 5B).

In the bench press, males recorded the highest 1RM values with combination and creatine, significantly higher than control: 98 ± 23, 83 ± 15 vs. 52 ± 12, *p* < 0.05 (Figure 5C). The same trend is maintained for females in the bench press, but the records are 2 times lower. The combination is at the top, while the creatine effect is slightly higher than the WP: 44 ± 9, 38 ± 10, and 32 ± 9 vs. 17 ± 5, *p* < 0.05 (Figure 5D).

The combination and creatine led to the highest 1RM values in male deadlifts: 103 ± 25, 93 ± 18 vs. 60 ± 15, *p* < 0.05 (Figure 5E). The same decreasing order is maintained in female deadlifts, but only combination and creatine are significantly higher than control: 74 ± 10 and 56 ± 7 vs. 37 ± 10. The effect of L-carnitine is substantially lower than the combination: 43 ± 10 vs. 74 ± 10, *p* < 0.05 (Figure 5F).

In exercises for biceps and triceps, males mostly benefitted from creatine; the corresponding 1RM values are considerably higher than controls: 53 ± 17, 60 ± 19 vs. 43 ± 6, 48 ± 6, *p* < 0.05 (Figure 5G,I). The combination was on top in pull-ups, followed by creatine: 15 ± 4, 11 ± 4 vs. 4 ± 3, *p* < 0.05 (Figure 5K). A similar trend is available for females in biceps, triceps, and pull-up exercises, but only combination and creatine are significantly higher than controls 36 ± 10, 47 ± 7, 6 ± 2 and 34 ± 4, 42 ± 5, 5 ± 2, vs. 20.6, 26 ± 6, 1 ± 1, *p* < 0.05 (Figure 5H,J,L).

#### 3.6.3. Age Group 51–60

Male and female participants aged 51–60 used only L-carnitine and WP. Their records are the lowest compared to other age groups, and no significant differences were observed compared to the controls. The daily diet can explain the differences between L-carnitine users in females (hyperprotein with 2501–3000 calories and 101–150 g proteins and vegetarian, with 1000–1500 calories and 51–100 g protein). The same aspects are available for men’s controls: the effects of daily protein consumption (>250 g) are supported by 1501–2000 calories (in a balanced diet) and 2501–3000 calories (in a hyperprotein diet), which led to a higher RT performance.

Generally, women’s RT performance is significantly lower than men’s in all three measurement times.

Thus, in back squat exercises, female vs. male 1RM values were as follows: 36 ± 14, 42 ± 16, 48 ± 18 vs. 69 ± 21, 81 ± 23, 93 ± 26, *p* < 0.05. In bench press, 19 ± 8, 24 ± 9, and 28 ± 11 vs. 58 ± 18, 69 ± 20, and 79 ± 24, *p* < 0.05, while in deadlift 37 ± 12, 44 ± 14, and 69 ± 16, *p* < 0.05.

In pull-ups, the following records were available: 2 ± 1, 3 ± 2, 4 ± 3 vs. 5 ± 3, 8 ± 5, and 10 ± 6, *p* < 0.05. In biceps the findings were as follows: 20 ± 5, 25 ± 7, and 28 ± 8 vs. 36 ± 8, 44 ± 9, and 51 ± 12, *p* < 0.05, while in triceps 25 ± 7, 31 ± 8, and 34 ± 10 vs. 40 ± 15, 50 ± 10, and 58 ± 13, *p* < 0.05.

### 3.7. Predictive Impact of Baseline Data and NS Consumption on RT Performance and Body Weight

To provide an overview of the entire study, the MANOVA test was performed to evaluate the effects of all baseline data and NS consumption on quantitative parameters measured at all three times (initially, after 4 weeks, and after 8 weeks). The results are displayed in Table 5.

Age, body weight status (provided by calculating BMI value), and NS type have a substantial impact on RT performance and body weight variance (*p* < 0.0001) of male participants. They are followed, in decreasing order, by daily diet-related factors: daily calories (*p* = 0.001), daily protein (*p* = 0.038), and diet type (*p* = 0.042).

Only NS type and body weight status significantly influence the female participants’ records (*p* < 0.0001). Daily protein consumption also has a considerable role (*p* = 0.001), followed by age (*p* = 0.014) and diet type (*p* = 0.028). Daily calorie intake is less influential for female gym-goers (*p* = 0.065, >0.05).

## 4. Discussion

The present study aimed to investigate the influence of protein and amino acid supplements (alone or in combination) on the RT performance of recreational gym goers. The project administrator carefully selected the participants based on inclusion/exclusion criteria. Therefore, the cohort is considerably heterogeneous in terms of age, gender, diet features, and NS consumption. Moreover, we aimed to assess the RT records of recreational gym-goers who consumed self-selected supplements. The reason for such a study design is the less investigated aspects of NS use in recreational athletes without medical recommendations. Additional data regarding dietary patterns were obtained from each participant through a face-to-face interview. The survey reported that WP, creatine, and L-carnitine are the most frequently used by recreational gym-goers in various doses. Their effects were evaluated initially by measuring the body weight, height, and 1RM for each RT exercise. The following measurements revealed the essential role of NS consumption in the RT record’s progressive improvement. The same manufacturers (known for their high-quality products) provided the dietary supplements used. The supplementation protocol was carefully established and followed to ensure the study’s repeatability and the accuracy of the measurements (Appendix A).

In our study, a higher proportion of women reported non-consumption of NSs compared to men, suggesting a lower reliance on supplementation among female participants. Among women who consumed NSs, WP and creatine intake were similarly preferred, L-carnitine was less frequent, and combined supplementation was the least common. In contrast, men exhibited a higher overall supplementation rate, with WP and creatine being less preferred. L-carnitine consumption was higher among men than women, indicating potential gender-based preferences or related to body weight status. The highest percentage of male participants opted for combination, reinforcing the trend of multi-component supplementation.

Moreover, age-related NS consumption is evident. Young adults (18–30 years) reported the highest use of NS, with significant differences between male and female gym-goers. This group preferred multi-component supplementation over single-supplement use. While these differences suggest potential correlations with training goals and physiological factors, these findings align with prior research on fitness center users, where men used WP and creatine supplements more frequently, particularly in the young adult category [67,68]. In contrast, participants aged 31–50 exhibited a more balanced distribution, preferring either creatine or whey protein and less opting for combined supplementation. The 50–50 age group displayed the highest non-consumption rate, revealing L-carnitine as the most favored option, likely due to its role in energy metabolism and muscle preservation. These trends align with previous findings that younger individuals prioritize strength and muscle gains, while older populations focus on overall health [49,69,70].

Other previous studies confirm our findings regarding the efficacy of these supplements in enhancing RT performance.

Esmark et al. demonstrated that early administration of an oral protein supplement after RTs is essential for developing skeletal muscle hypertrophy in older adults [71]. Other researchers have found that post-game high-protein intake may improve the recovery of football-specific performance during a congested game fixture [72]. Cermak et al. confirmed the efficacy of protein supplementation in augmenting the adaptive response of skeletal muscle to prolonged resistance-type exercise training in both younger and older populations [73]. Another study highlighted that enriching a protein drink with leucine increases muscle protein synthesis after resistance exercise training in young and older men [74]. Hansen et al. reported that the intake of whey protein hydrolysate (0.3 g/kg) before and after exercise sessions increases endurance performance and recovery in 18 elite orienteers during a 1-week training camp with 13 exercise sessions [75,76,77].

The high RT performance of gym-goers with creatine supplementation was previously confirmed. Galvan et al. conducted a trial involving 13 healthy and physically active adults divided into four groups, each supplemented with a different dose of creatine (1.5 g, 3 g, and 5 g/day), aiming to assess the dose-dependent effects on safety and exercise performance rates. The authors concluded that a dose of up to 3 g/day is safe and effective regarding changes in strength and body composition [78]. Yáñez-Silva et al. highlighted muscle power increase in young elite football players supplemented with low creatine doses (0.03 g/kg/day) for 14 days [79]. On the other hand, a study on nineteen healthy recreational male bodybuilders found no significant differences in body composition and strength between creatine administration before and after RT [80]. A similar observation was reported in another study involving 22 healthy older adults (9 males, 13 females, aged 50–64 years) who performed strength training over 12 weeks, 3 days a week [81]. Rawson and Wolek’s meta-analysis revealed that creatine supplementation and resistance training resulted in an average increase of +8% and +14% in 1RM performance compared to the placebo groups [82].

L-carnitine tartrate supplementation has a beneficial effect on markers of post-exercise metabolic stress and muscle damage. Spiering et al. demonstrated that 1–2 g L-carnitine effectively mitigated various markers of metabolic stress and muscle soreness in athletes [36], while 3–4 g taken before physical exercise prolonged exhaustion [37].

The substantial RT performance induced by the triple combination consumption throughout the study period indicates a possible synergistic effect of WP, creatine, and L-carnitine in optimizing muscle function. A previous study reported the benefits of L-carnitine combined with creatine and leucine on functional muscle strength in healthy older adults [38]. Another study examined the effects of whey protein and creatine supplementation compared to WP consumption alone on body composition and performance variables in 17 resistance-trained young women [83]. After 8 weeks of training, Wilborn et al. observed that all performances increased, but no significant differences were recorded between WP and WP + creatine supplementation [83]. Similar findings were reported by Collins et al. in the RT of elderly individuals with frailty [84]. However, our study reveals that the triple combination (WP + creatine + L-carnitine) led to significantly higher RT performance than the component alone when it is associated with a balanced diet, a hyperprotein diet, and a vegetarian diet. These findings fit the previous observation that protein supplementation is linked to time spent exercising and high-protein-content foods [85].

Our results show that gym performance decreases with age, and 51–60 aged gym-goers obtained the lowest performance. The observed patterns align with the natural physiological decline in muscle mass, neuromuscular efficiency, and recovery capacity associated with aging. It suggests that individuals over 50 may experience a more pronounced decrease in lower-body strength, and the slow process of sarcopenia is thought to begin [86]. Age-related factors, such as decreased anabolic hormone levels like testosterone and growth hormone, may reduce muscle strength and recovery efficiency [70,87]. It is well-documented that many biological changes that come with aging, such as modifications to the structure and function of most organs, such as the heart, skeletal muscles, arteries, and brain, can account for this. More broadly, exercise performance decreased as age increased [88,89]. Although strength training can mitigate these effects, the results suggest that younger individuals retain a distinct advantage in absolute strength levels. The muscle mass maintenance involves a balance between muscle protein synthesis (MPS) and muscle protein breakdown (MPB). The synergistic action of exercise and amino acid/protein ingestion increases MPS [90]. The most recent American College of Sports Medicine position recommends a protein intake of 1.2–1.7 g/kg/day for endurance- and resistance-trained gym-goers [91]. Athletes can consume protein before, during, and after exercise [71,92,93,94]. There are different theories as to which period promotes an optimum adaptation. Still, in the case of resistance exercise, almost all of them relate to the ability of the protein to provide amino acid precursors to either support MPS or inhibit MPB [71,95,96]. Despite declining performance with age, resistance training helps maintain muscle function and overall physical health [97,98,99,100]. Numerous studies investigated protein and amino acid supplementation associated with RT benefits in older people [12,18,20,23,61,74,81,84,101,102,103,104,105,106,107,108,109,110,111]

Gender-related differences were evident in all exercises, with men consistently outperforming women. This disparity is likely attributed to physiological factors such as higher muscle mass, greater testosterone levels, and differences in neuromuscular efficiency. However, both genders showed progressive improvements in strength over the training period, emphasizing the effectiveness of resistance training regardless of baseline differences. Participants aged 18–50 are of two BMI types (NW and OW); at the same time, they form a heterogeneous group, including controls and all NS consumers. Therefore, the performances recorded in W8 are compared in this age range. Generally, gym performance decreased with age, particularly in the 18–29 to 30–49 age groups; moreover, males’ records were usually higher than those of females. These aspects were analyzed based on the most significant results recorded in W8 for males and females of both age groups, including those with NNSUs and all NSUs (Figure 5).

The final measurements (W8) indicate that combination consumption for both sexes resulted in the most significant improvement in gym performance after 8 weeks in the back squat, bench press, deadlift, biceps, and triceps, compared to control (*p* < 0.05). Creatine supplementation in men produced performance levels similar to those of combination intake. In women, L-carnitine supplementation led to a remarkable performance improvement compared to the control (*p* < 0.05), while WP recorded the lowest performance. Creatine consumption was associated with minimal results for most women, showing a slight improvement compared to control (*p* > 0.05). Our findings are similar to those from other studies, reporting that creatine did not reach statistical significance over a placebo in females. There was a trend for more significant upper-body strength gains from creatine in males than in females [22]. The creatine-supplemented males experienced a more substantial improvement in upper-body strength than control (*p* < 0.05), even if creatine monohydrate is one of the most used ergogenic supplements in high-intensity and/or strength-based activities (e.g., RT). However, creatine and L-carnitine had the lowest effects on pull-up exercises in males and females, respectively. Moreover, all performances substantially vary with NS type, age, gender, diet type, daily proteins, and daily calorie consumption (*p* < 0.05).

The lowest gains of the control group suggest that resistance training alone may not be sufficient to maximize strength development within this timeframe.

Future research with controlled dietary intake and larger sample sizes must further elucidate each supplement’s contributions to muscle strength and resistance training adaptations. It should investigate the effects of training history, supplementation, and recovery strategies on mitigating age-related declines in strength. Additionally, longitudinal studies examining individual strength progression over time may provide deeper insights into optimizing resistance training programs for older populations.

### Strength and Limitations

Our study offers complex insights, analyzing the effects of self-selected supplements (WP, creatine, and L-carnitine) consumed alone and in a triple combination on RT performances in a heterogeneous group of recreational gym-goers. The complex analysis of RT records evolution for 8 weeks involved 1RM measurement correlation with baseline data (age, gender, dietary patterns, and body weight status). The participants’ heterogenicity is considerable and allows us to evaluate the variation in RT records with age, gender, body weight status, and diet patterns. Moreover, the extensive statistical analysis correlated RT performance with other influential factors, showing statistically significant co-variances and predictive status. Therefore, all hypotheses were verified.

Almost three-quarters of gym-goers consume protein and amino acid supplements. Young adults (18–30 years) are the main consumers of these supplements; most prefer the triple combination. Using various statistical tools, our study revealed the substantial impact of NS consumption, age, gender, body weight status, daily diet type, daily calories, and protein consumption on RT performances.

Several limitations must be acknowledged as follows:Various elements may affect strength performance, including training history, genetic predisposition, and recovery strategies; however, these individual differences were not thoroughly controlled.The study only examined short-term impacts (8 weeks). A more extended intervention period would be needed to assess the durability of strength gains.A content analysis of the supplements was not performed; neither sample was collected nor preserved for a long time. However, each NS was carefully selected considering the manufacturer’s reputation and the fabrication technology; the product labels and the available Appendix A accessed on the supplier websites were the only sources regarding its content and quality.While most supplements are administered at the gym location and supervised by qualified trainers, nutritional supplementation is monitored only through individual statements in male participants who take the first dose of WP at home in the morning.Due to the numerous participants and the group’s heterogenicity, the diets of recreational gym-goers were not monitored; therefore, the influence of dietary patterns on RT performance was not evaluated.

## 5. Conclusions

Despite all limitations, the present study confirms that resistance training and protein and amino acid supplementation benefit males and females across all age and BMI categories. Our findings show that whey protein, creatine, and L-carnitine, consumed in combination, are mostly preferred in the self-decided supplementation of recreational gym-goers aged 18–50. At the same time, L-carnitine was used most by males and females aged 50–60. The most significant records during 8 weeks of resistance training were obtained by triple combination consumers, suggesting that all three components act synergistically on muscle strength and recovery. Our data show that male participants performed the highest through creatine supplementation, while female gym-goers benefited significantly from L-carnitine consumption. Gender, age, NS type, body weight status, and diet features significantly influence strength outcomes—the RT performance was higher in male participants and progressively decreased with age; L-carnitine consumers lose weight during resistance training. Future research should explore the medical nutrition intervention in daily diet and supplementation strategy in recreational gym-goers to maintain normal body weight and optimize strength performance during personalized resistance training protocols.

## Figures and Tables

**Figure 1 sports-13-00182-f001:**
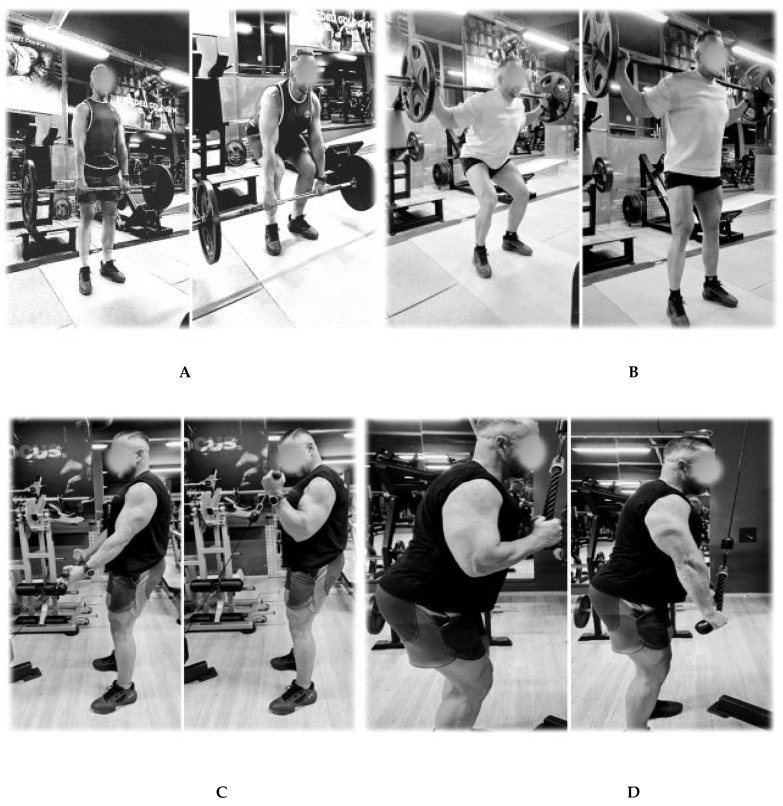
Pictograms of the RT exercises: (**A**) deadlift, (**B**) back squat, (**C**) biceps, (**D**) triceps, (**E**) pull-ups, and (**F**) bench press.

**Figure 2 sports-13-00182-f002:**
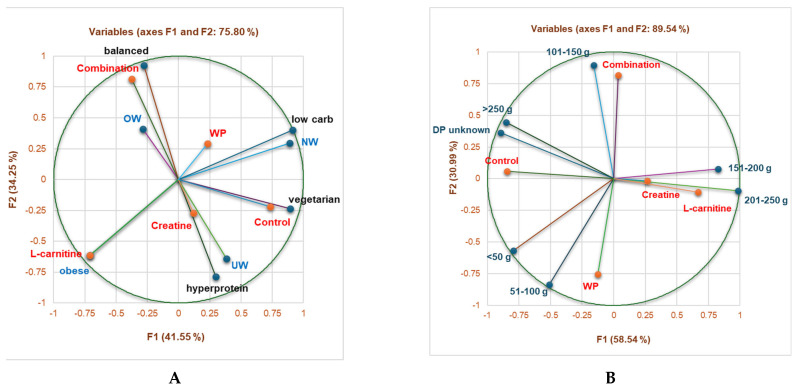
Correlations between NS consumption and (**A**) BMI status and diet type, (**B**) daily protein (g), and (**C**) daily calories. Statistical tool: Principal Component Analysis through Pearson correlation.

**Figure 3 sports-13-00182-f003:**
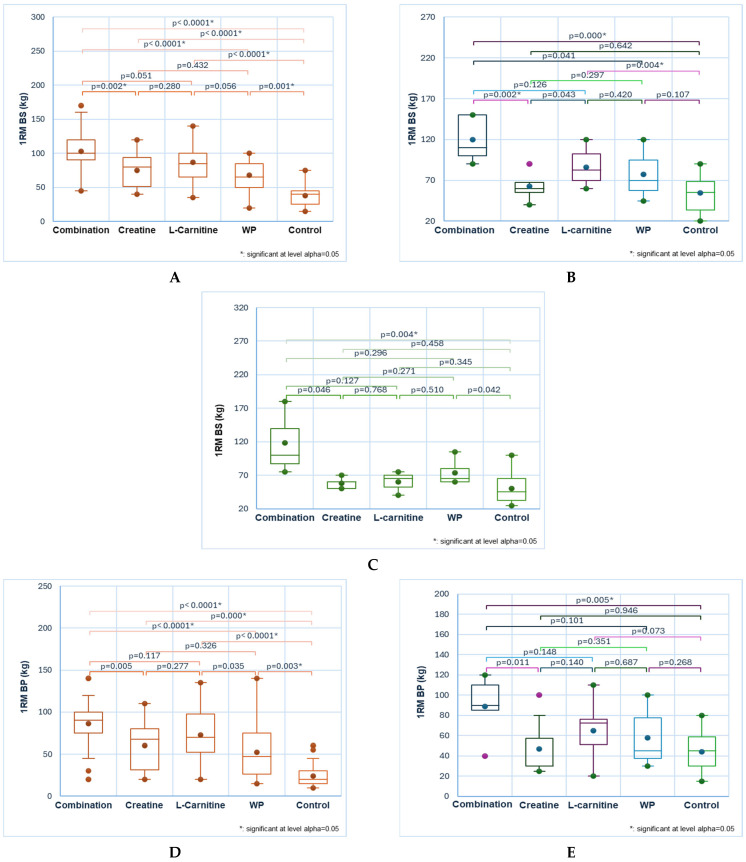
The NS influence on RT performance in gym-goers with different diets: balanced (**A**,**D**,**F**,**I**,**J**,**K**), hyperprotein (**B**,**E**,**G**), vegetarian (**C**,**H**), and low-carb (**L**). 1RM back squat (**A**–**C**); 1RM bench press (**D**,**E**); 1RM deadlift (**F**–**H**); 1RM biceps (**I**); 1RM triceps (**J**); pull-ups (**K**,**L**); Bonferroni corrected significance level = 0.005. Statistical tool: Kruskal–Wallis test. Different colors were used to facilitate their differentiation: red for a balanced diet, mixed color for a hyperprotein diet, green for a vegetarian diet, and purple for a low–carb diet.

**Figure 4 sports-13-00182-f004:**
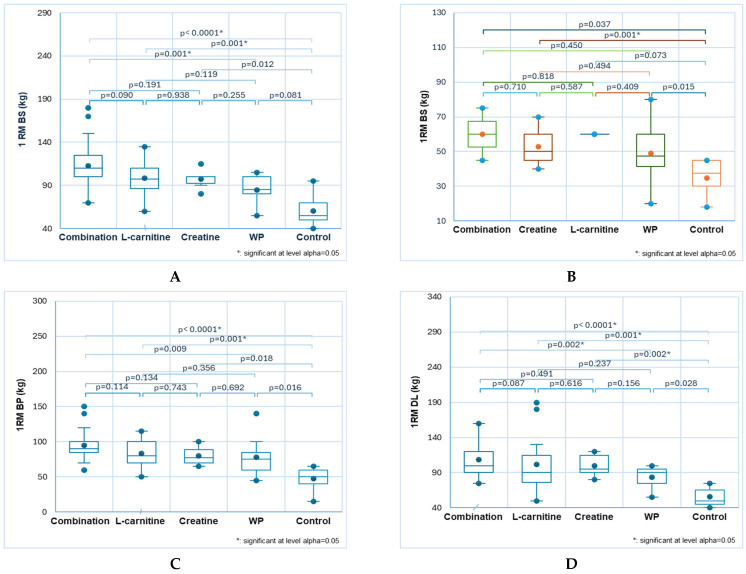
The NS influence on RT performance in gym-goers aged 18–30: males (**A**,**C**–**F**,**H**) and females (**B**,**G**,**I**); 1RM back squat (**A**,**B**); 1RM bench press (**C**); 1RM deadlift (**D**); 1RM biceps (**E**); 1RM triceps (**F**,**G**); 1RM pull-ups (**H**,**I**). Bonferroni corrected significance level = 0.005. Statistical tool: Kruskal–Wallis test. Different colors were used to facilitate their differentiation: blue for males and mixed colors for females.

**Figure 5 sports-13-00182-f005:**
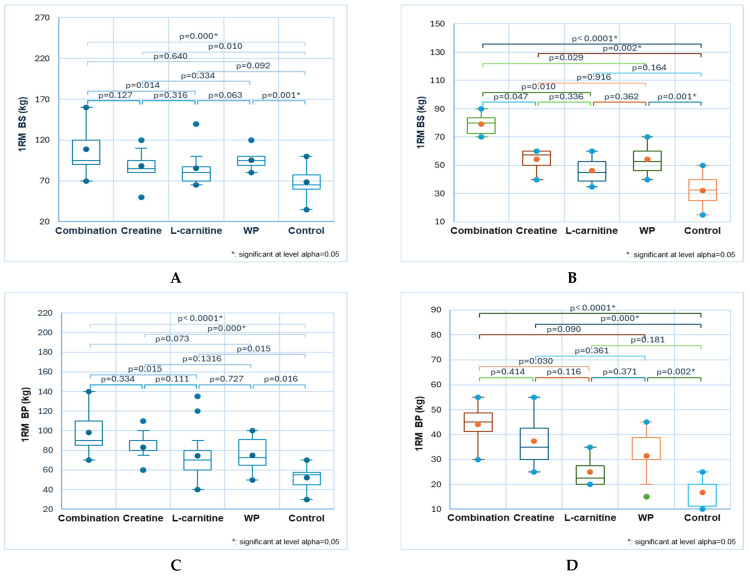
The NS influence on RT performance in gym-goers aged 31–40: males (**A**,**C**,**E**,**G**,**I**,**K**) and females (**B**,**D**,**F**,**H**,**J**,**L**); 1RM back squat (**A**,**B**); 1RM bench press (**C**,**D**); 1RM deadlift (**E**,F); 1RM biceps (**G**,**H**); 1RM triceps (**I**,**J**); 1RM pull-ups (**K**,**L**). Bonferroni corrected significance level = 0.005. BS—back squat, BP—bench press, DL—deadlift. Statistical tool: Kruskal–Wallis test. Different colors were used to facilitate their differentiation: blue for males and mixed colors for females.

**Table 1 sports-13-00182-t001:** Sociodemographic and baseline characteristics of the study participants.

Parameter	Total group		Female		Male		*p*-Value
	*n*	%	*n*	%	*n*	%	
Total	218	100	86	39.45	132	60.55	<0.05
Age (years)							
18–30	113	51.83	40	46.51	73	55.30	<0.05
31–50	98	44.95	42	48.84	56	42.42	<0.05
51–60	7	3.21	4	4.65	3	2.27	<0.05
Body Weight Status							
NW	112	51.38	65	75.58	47	35.61	<0.05
OW	98	44.95	16	18.60	82	62.12	<0.05
UW	6	2.75	5	5.81	1	0.76	<0.05
obese	2	0.92	0	0.00	2	1.52	<0.05
Daily Diet							
balanced	130	59.63	46	53.49	84	63.64	<0.05
hyperprotein	46	21.10	19	22.09	27	20.45	<0.05
low carb	15	6.88	6	6.98	9	6.82	<0.05
vegetarian	27	12.39	15	17.44	12	9.09	<0.05
Daily Calories							
1000–1500	34	15.60	17	19.77	17	12.88	<0.05
1501–2000	71	32.57	26	30.23	45	34.09	<0.05
2001–2500	42	19.27	16	18.60	26	19.70	<0.05
2501–3000	43	19.72	17	19.77	26	19.70	<0.05
3001–3500	13	5.96	4	4.65	9	6.82	<0.05
<1000	1	0.46	1	1.16	0	0.00	<0.05
>3500	1	0.46	0	0.00	1	0.76	<0.05
DC unknown	13	5.96	5	5.81	8	6.06	<0.05
Daily Proteins (grams)							
101–150 g	55	25.23	18	20.93	37	28.03	<0.05
151–200 g	34	15.60	13	15.12	21	15.91	<0.05
201–250 g	29	13.30	9	10.47	20	15.15	<0.05
51–100 g	44	20.18	23	26.74	21	15.91	<0.05
<50 g	2	0.92	2	2.33	0	0.00	<0.05
>250 g	18	8.26	8	9.30	10	7.58	<0.05
DP unknown	36	16.51	13	15.12	23	17.42	<0.05
NS Consumption							
Control	53	24.31	31	36.05	22	16.67	<0.05
Combination	48	22.02	8	9.30	40	30.30	<0.05
WP	42	19.27	21	24.42	21	15.91	>0.05
Creatine	38	17.43	19	22.09	19	14.39	>0.05
L-carnitine	37	16.97	7	8.14	30	22.73	<0.05

*n*—participants’ number (frequency), %—percentage (relative frequency), NS—nutritional supplements, NW—normal weight, OW—overweight, UW—underweight; Control—no NS consumers; WP—whey protein; statistical significance: *p* < 0.05.

**Table 2 sports-13-00182-t002:** RT performance records of all participants and covariances.

Measurement	W0	W4	W8
1 RM Back Squat (kg)			
Min	10	15	15
Max	140	160	180
Mean	56	66	75
SD	25	28	32
Covariance	*p*-Value		
Gender	<0.0001	<0.0001	<0.0001
NS type	<0.0001	<0.0001	<0.0001
Body weight status	<0.0001	<0.0001	<0.0001
Daily calorie	0.037 *	0.052	0.109
Daily proteins (g)	0.040 *	0.052	0.048 *
1RM Bench Press (kg)			
Min	10	10	10
Max	120	130	150
Mean	43	51	59
SD	24	28	32
Covariance	*p*-Value		
Gender	<0.0001	<0.0001	<0.0001
NS type	<0.0001	<0.0001	<0.0001
Body weight status	<0.0001	<0.0001	<0.0001
1RM Deadlift (kg)			
Min	10	15	15
Max	140	160	190
Mean	54	65	74
SD	22	26	31
Covariance	*p*-Value		
Gender	<0.0001	<0.0001	<0.0001
NS type	<0.0001	<0.0001	<0.0001
Body weight status	<0.0001	<0.0001	<0.0001
Age (years)	0.234	0.066	0.013 *
1RM Biceps (kg)			
Min	10	10	10
Max	60	70	85
Mean	29	36	42
SD	11	13	16
Covariance	*p*-Value		
Gender	<0.0001	<0.0001	<0.0001
Body weight status	<0.0001	<0.0001	<0.0001
NS type	<0.0001	<0.0001	<0.0001
Daily proteins (g)	0.038 *	0.017 *	0.013 *
Diet type	0.139	0.075	0.048 *
1RM Triceps (kg)			
Min	5	10	15
Max	60	80	90
Mean	34	43	49
SD	11	13	17
Covariance	*p*-Value		
Gender	<0.0001	<0.0001	<0.0001
Body weight status	<0.0001	<0.0001	<0.0001
NS type	<0.0001	<0.0001	<0.0001
Daily proteins (g)	0.012 *	0.005 *	0.003 *
Diet type	0.154	0.078	0.025 *
Pull-Ups (count)			
Min	0	0	0
Max	15	30	38
Mean	4	6	8
SD	3	5	6
Covariance	*p*-Value		
Gender	<0.0001	<0.0001	<0.0001
NS type	<0.0001	<0.0001	<0.0001
Diet type	0.018 *	0.022 *	0.018 *
Body weight status	0.041 *	0.105	0.125
Age (years)	0.228	0.026 *	0.012 *

* statistically significant, *p* < 0.05; NS—nutritional supplement; SD—standard deviation; statistical tool: Fischer’s F-test.

**Table 3 sports-13-00182-t003:** The impact of NS consumption on body weight after 8 weeks.

NS Type	Body Weight Status—Measurement Time	Minimum Weight	Maximum Weight	Mean	Std. Deviation	** p*-Value
Combination	NW—W0	46	85	70	10	<0.05
	NW—W8	50	88	72	10	
	OW—W0	68	132	91	13	
	OW—W8	65	130	92	13	
Creatine	NW—W0	49	90	66	11	<0.05
	NW—W8	53	93	68	10	
	OW—W0	70	108	89	11	
	OW—W8	74	114	92	12	
WP	NW—W0	47	92	67	11	<0.05
	NW—W8	49	97	67	12	
	OW—W0	71	124	92	13	
	OW—W8	71	120	93	12	
Control	NW—W0	45	88	59	10	<0.05
	NW—W8	47	83	58	10	
	OW—W0	56	128	87	19	
	OW—W8	57	123	85	19	
L—carnitine	NW—W0	50	81	67	11	<0.05
	NW—W8	49	80	64	10	
	OW—W0	66	113	90	13	
	OW—W8	64	106	86	12	
	obese—W0	130	130	130	0	
	obese—W8	122	124	123	1	

* significant differences between NW and OW or obese. NS—nutritional supplements; NW—normal weight; OW—overweight; Control—no NS consumers; WP—whey protein; W0—initial measurement; W8—final measurement (after 8 weeks). Statistical significance: *p* < 0.05.

**Table 4 sports-13-00182-t004:** Body weight status, diet, and NS consumption in males and females of different ages.

Age	18–30		31–50		51–60	
Sex	F	M	F	M	F	M
Body weight status						
NW	75.00	46.58	78.57	23.21	50.00	0.00
OW	15.00	49.32	19.05	76.79	50.00	100.00
UW	10.00	1.37	2.38	0.00	0.00	0.00
obese	0.00	2.74	0.00	0.00	0.00	0.00
Diet type						
balanced	47.50	63.01	64.29	64.29	0.00	66.67
hyperprotein	25.00	17.81	19.05	23.21	25.00	33.33
low carb	5.00	9.59	7.14	3.57	25.00	0.00
vegetarian	22.50	9.59	9.52	8.93	50.00	0.00
NS type						
Combination	5.00	42.47	14.29	16.07	0.00	0.00
Control	40.00	12.33	33.33	19.64	25.00	66.67
Creatine	27.50	8.22	19.05	23.21	0.00	0.00
L-carnitine	2.50	19.18	9.52	26.79	50.00	33.33
WP	25.00	17.81	23.81	14.29	25.00	0.00

All values are expressed as relative frequencies.

**Table 5 sports-13-00182-t005:** Predictive effects of baseline data and NS consumption on male and female participants.

Impact	Age (Years)	Body Weight Status	Diet Type	Daily Calories	Daily Proteins (g)	NS Type
Males						
*p*-value	<0.0001	<0.0001	0.042	0.001	0.038	<0.0001
Females						
*p*-value	0.014	<0.0001	0.028	0.065 *	0.001	<0.0001

* not statistically significant (*p* > 0.05); NS—nutritional supplement; Statistical tool: MANOVA test.

## Data Availability

The original contributions presented in the study are included in the article; further inquiries can be directed to the first author and corresponding authors.

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
