# Peer review of "The Impact of Protein and Amino Acid Supplementation on Muscular Strength and Endurance in Recreational Gym-Goers During 8-Week Resistance Training"

_sports, 2025, doi:10.3390/sports13060182_

Round 1
Reviewer 1 Report
Comments and Suggestions for Authors
sports-3625104
The present manuscript investigated the supplements consumption in gym-goers. This is an interesting research subject, and I will present below my evaluation point-by-point.
I suggest a long and detailed review of the text regarding the use of acronyms. There are many and, in my opinion, they are unnecessary, as they make it difficult to understand the text. I see no need to create an acronym for Creatine (CS), L-carnitine (LcS), whey protein (PS), Female (F), Male (M). Furthermore, there are different acronyms for the same term [e.g. non-supplemented group (NNS) and NNSUs]. There are acronyms that are not explained at any point in the text. What are PSUs? Acronyms that are used in rare situations should not be created (ex. EAA). Finally, when creating an acronym, the written text should come first [e.g. one maximum repetition (1MR)]. Please review it, as there are errors.
Many of the results presented in the Abstract only present percentages. I recommend that authors include the p-value with each result. There are 10 keywords, some of which are redundant. I recommend that authors review them.
The text needs to be grammatically revised. There are many punctuation errors, sentences without periods, for example. Before submitting the next version, the authors should pass the text to a proofreader.
The introduction is very long and repetitive. There are 9 paragraphs that can be summarized, e.g. the first and second paragraphs can be unified. There is a lot of information that is clear in the literature and, therefore, does not need to be presented in such an extensive manner, such as the function of whey protein, amino acids, creatine and L-carnitine. The eighth paragraph repeats information that was already presented in the second. Finally, add the hypothesis at the end of the introduction.
The methods are difficult to understand. I recommend that the authors create a Subhead called “Experimental approach”, in which the authors will describe how the study was organized. E.g. the authors report the moment W0, but it is not possible to understand what W0 is if the authors do not explain first that the study is divided into several measurement moments. Please redo it so that it is clear to the readers.
It is not necessary to create a Table just to present the inclusion and exclusion criteria. A Table is justified when it will replace a large amount of text, which is not the case here. How relevant is Table 2 to the text? In my understanding, the authors can carry out the entire study without it, I suggest removing it.
What are the inclusion and exclusion criteria? Did all participants complete the study? Please make this subhead clearer. The study has a large sample size, which is positive, however, it is necessary to present a sample representativeness calculation.
What was the procedure for checking whether the participants had intake the supplements? This procedure is essential for a study with this type of design and the description is not clear enough. Furthermore, the authors do not make any indication of control over the participants' diet. This is a serious error that brings an important bias to the study. I suggest that the authors add this limitation at the end of the discussion.
The training Figures are completely unnecessary, they are widely known and do not bring any innovation, I suggest removing them all.
What method is used to measure 1MR? There is no clear description about it.
Was any control performed regarding the use of anabolic steroids?
Figures 2, 3 and 4 could be better presented, there is no need to create letters, the authors can write in the Figures what each one represents (e.g. 2nd – Male 18-29 years). The authors should keep in mind that when writing the text, they should do so in a way that helps the reader understand. Are there no statistical differences in Figure 2, 3 and 4 or did the authors not perform statistical procedures? Please clarify.
The data from Table 4 can be presented in Table 3.
In the discussion, the first paragraph should provide a qualitative analysis of the main results of the study.
Authors should completely separate the results from the discussion. Figures 5 and 6 should be in the results.
There is a lot of unnecessary information in the discussion, the authors repeat many of the results previously presented, as noted in the first paragraph of Subhead 4.1. Please remove it.
The text is interesting, but there are many errors and inconsistencies. The authors should make an effort and address the suggested corrections.
Author Response
Dear Reviewer 1,
The authors are grateful for your time, willingness, and accurate comments, aiming to improve the quality of our MS.
All changes are revealed in the revised version, and all responses are made point by point.
It was an extensive revision; that is the reason for the supplementary time requested; the authors would be grateful to know they succeeded.
Please find the corresponding response in the attachment.

Reviewer 2 Report
Comments and Suggestions for Authors
This study aimed to collect data on the consumption of nutritional supplements among recreational gem-goers and examine their influence on gym performance during an 8-week resistance exercise training program. While I commend the authors on their efforts and the amount of data collected, there are significant issues in the study that should be addressed before the paper can be properly evaluated for publication. I believe one of the major issues is that the purpose of the study is not well justified in the introduction and the overall purpose is unclear and weak. The methods are not clearly stated and are not written in a way where the study could be replicated. The statically analyses section is also not clear, which makes the results difficult to interpret. The results themselves are presented in an unintuitive manner. The discussion is mostly an expanded version of the results and does not communicate to the reader the importance of the findings and how they advance the field forward. Below are more detailed comments about these issues.
Introduction
Although the introduction provides a comprehensive review of several popular nutritional supplements, it too long and never really identifies a significant gap or issue. The main focus of the research is to investigate the effects of whey protein, creatine, and carnitine, both in isolation and combination, on resistance training performance in a diverse population. However, there are several issues. The authors already provide ample evidence in the introduction that these supplements are effective, to various degrees, for improving physical performance in a variety of studies. Since these studies are conducted with different individuals in different parts of the world, the participants across all studies would be diverse. However, the authors only included individuals from two gyms, that appeared to be relatively close together, within the same city, which undermines one of the main objectives. Furthermore, there are many studies that have already investigated the combination of whey protein and creatine. Additionally, there is no justification as to why only these three supplements are the focus. Beta-alanine, citrulline and others are also very popular supplements. So why only whey protein, creatine, and carnitine? Overall, the introduction focuses too much on reviewing the literature on the supplements and fails to demonstrate an issue or gap in the existing literature to justify the need for this study.
Methods
What was the rationale for the supplement dosages, other than that was the manufacturer’s recommended dosages?
Authors did not collect any nutritional intake date during the study, this is a huge limitation and there should be ample justification.
Did the authors take post training weight? If not, then how is the potential change in body weight accounted for when interpreting changes in strength performance measurements?
Was any information collected on the pre-study supplement dosages of the participants?
What was the rationale for the protein supplementation? Why are males provided with two dosages and women one?
The resistance training protocol is unclear. For example, on line 238 the authors state 5 sets were performed, but then on line 248, it appears as though only 3 sets were performed. I also do not understand how the load was progressively increased during the training program. For example, on lines 239-240 it states that weekly training loads were increased by 2-10% based on the individual’s strength, but then on line 251 it states that 1RM was “augmented” every 4 weeks based on the muscle strength gained. Does that mean loads were increased every week and then readjusted after the 4th week? Sinc participants trained 3 times per week, did they perform all exercises each session? If not, how was that handled?
Statistical analysis
I am not clear what a “Fischer test” is. Do the authors mean a Fisher’s exact test? If so, then then that test is not used for 2-sample comparisons of variances. I assume the authors are referring to some type of t-test or f-test (maybe an ANOVA), since they mention comparisons of variances; however, it is not clear and requires more detail to ensure the analysis was done correctly and accounted for repeated measured (assuming that some of the analyses compared the groups and changes in strength) when appropriate.
The variables that were used for the “Fischer test” should be specified.
For lines 263-265 the sentence “Additionally, gym performance was correlated with NS consumption, and the analysis was conducted while considering the age, sex, and BMI type of all participants” is unclear. How was gym performance defined? What criteria was used to determine the gym performance and NS were correlated? Everything is correlated to some degree, so was the correlation significant? And what is the importance of this statement? How did it influence the analysis? What is meant by “the analysis was conducted considering…”? Were these variables used as covariates? If so, what justification do the authors have for including these in the analysis as covariates?
There appear to be many multiple comparisons. What method was used to correct for this?
Did the authors consider baseline strength between groups as a covariate? There appears to be a large degree of variation that may confound the results.
There is no justification for the sample size.
Based on the potential changes to the analysis, the results may be significantly influenced.
Results
Line 316 “Body weight is another factor that substantially influences gym performance” is not supported by the presented data and should be justified or removed.
I understand the authors have a lot of data to present; however, figure 3 is very difficult to interpret because the x axis shows groups and not time and the use of lines makes it appear as though participants were in more than one group. This makes it very difficult to determine which groups improved. There should be some justification for the selected age categories (why every 10 years).
Discussion
The discussion mostly is an extension of the results rather than interpreting what the results mean and in the context of the previous literature.
Lines 495-499 may need to be reconsidered after evaluating the statistical analysis section.
Lines 500-506 seem out of place and maybe belong in the methods. I do not understand what they add to the discussion.
Line 538 “likely due to its role in energy metabolism and muscle preservation” How do the authors know that the participants knew what carnitine was used for? Did the authors survey the participants on their knowledge of the supplements they were consuming?
Author Response
Dear Reviewer 2,
The authors are grateful for your time, attention, and accurate comments, which aim to improve the quality of our manuscript. The revision was complex, with many data added and analyzed; that is the reason for our extended period.
Please find attached the authors response.

Round 2
Reviewer 1 Report
Comments and Suggestions for Authors
In the current version of the manuscript, it is possible to notice great improvements by the authors in several points. Those that were not revised had an acceptable justification. Therefore, I am in favor of publishing the current version.
Reviewer 2 Report
Comments and Suggestions for Authors
The authors have made adequate revisions to the paper to make it acceptable for publication.